# A Novel Fibromodulin Antagonist Peptide RP4 Exerts Antitumor Effects on Colorectal Cancer

**DOI:** 10.3390/pharmaceutics15030944

**Published:** 2023-03-14

**Authors:** Ting Deng, Yibo Hou, Gaoyang Lin, Chunyan Feng, Kewei Liu, Wenke Chen, Wei Wei, Laiqiang Huang, Xiaoyong Dai

**Affiliations:** 1Precision Medicine and Healthcare Research Center, Center for Biotechnology and Biomedicine, Shenzhen Key Laboratory of Gene and Antibody Therapy, State Key Laboratory of Chemical Oncogenomics, State Key Laboratory of Health Sciences and Technology, Tsinghua-Berkeley Shenzhen Institute (TBSI), Institute of Biopharmaceutical and Health Engineering, Shenzhen International Graduate School, Tsinghua University, Shenzhen 518055, China; 2Institute of Biopharmaceutical and Health Engineering, Shenzhen International Graduate School, Tsinghua University, Shenzhen 518055, China; 3Department of Chemistry, Tsinghua University, Beijing 100084, China; 4Peking University Shenzhen Hospital, Shenzhen 518036, China

**Keywords:** fibromodulin, colorectal cancer, metastasis, tumor microenvironment, AKT signaling pathway, Wnt/β-catenin signaling pathway

## Abstract

Colorectal cancer (CRC) is the leading cause of cancer-related deaths worldwide. Fibromodulin (FMOD) is the main proteoglycan that contributes to extracellular matrix (ECM) remodeling by binding to matrix molecules, thereby playing an essential role in tumor growth and metastasis. There are still no useful drugs that target FMOD for CRC treatment in clinics. Here, we first used public whole-genome expression datasets to analyze the expression level of FMOD in CRC and found that FMOD was upregulated in CRC and associated with poor patient prognosis. We then used the Ph.D.-12 phage display peptide library to obtain a novel FMOD antagonist peptide, named RP4, and tested its anti-cancer effects of RP4 in vitro and in vivo. These results showed that RP4 inhibited CRC cell growth and metastasis, and promoted apoptosis both in vitro and in vivo by binding to FMOD. In addition, RP4 treatment affected the CRC-associated immune microenvironment in a tumor model by promoting cytotoxic CD8^+^ T and NKT (natural killer T) cells and inhibiting CD25^+^ Foxp3^+^ Treg cells. Mechanistically, RP4 exerted anti-tumor effects by blocking the Akt and Wnt/β-catenin signaling pathways. This study implies that FMOD is a potential target for CRC treatment, and the novel FMOD antagonist peptide RP4 can be developed as a clinical drug for CRC treatment.

## 1. Introduction

Colorectal cancer (CRC) is one of the leading causes of cancer-related deaths in both sexes worldwide. CRC decreases patient well-being and life expectancy. According to the GLOBOCAN estimation of all-cancer incidence and mortality, CRC is the third and second most common cancer with incidence and mortality rates of 10.0% and 9.4%, respectively [1,2]. The incidence rate of CRC in transitioned countries is approximately four-fold higher than that in transitioning countries. Therefore, CRC is considered a marker of socioeconomic development [3].

The tumor microenvironment (TME) reciprocates with tumor cells to support cancer cell survival, local invasion, and metastatic dissemination. The TME generally comprises immune cells, stromal cells, blood vessels, and extracellular matrix [4,5]. As critical components of the TME, immune cells play dual roles in tumor growth, either by suppressing or promoting tumor growth [6]. Cytotoxic T cells (CD8^+^ T) have strong anti-tumor abilities and directly kill cancer cells by detecting abnormal tumor antigens expressed on cancer cells [7]. In addition, CD8^+^ T cells secrete IFN-γ to suppress angiogenesis [8]. In contrast, regulatory T cells (Tregs) promote cancer cell survival by secreting growth factors and interacting with the stromal cells in the microenvironment [9]. Furthermore, IL-2 secreted by Tregs assists cancer cell survival by modulating natural killer (NK) cell homeostasis and function. In addition to immune cells, extracellular components secreted by various cells in the TME also promote tumor cell dissemination [10,11]. Matrix metalloproteinases (MMPs) are classical proteases that degrade various extracellular matrix proteins and induce epithelial–mesenchymal transition (EMT) to promote tumor cell migration and invasion [12]. Moreover, canonical Wnt signaling (β-catenin-dependent) drives EMT to promote CRC metastasis [13].

Fibromodulin (FMOD) is a secreted proteoglycan belonging to the family of small interstitial leucine-rich repeat proteoglycans (SLRPs), which are essential regulators of extracellular matrix assembly and cell signaling [14]. FMOD is initially regarded as a collagen-binding protein that is widely present in connective tissues including skin, cartilage, and tendons [15]. Incipient investigations confirm that FMOD, as well as lumican, which is another member of the SLRPs family, regulates extracellular matrix (ECM) organization and structure in connective tissues by binding to collagen I and II [16,17]. However, accumulating evidence demonstrates a complex relationship between FMOD and tumorigenesis. In experimental carcinoma, FMOD positively modulates collagen assembly to maintain a dense collagen scaffold and fluid balance, further protecting cancer cells from anti-cancer drugs [18]. Moreover, FMOD is overexpressed in multiple cancers, including lung cancer, prostate cancer, and chronic lymphocytic leukemia, and is a potential biomarker for B-cell acute lymphoblastic leukemia [19,20,21]. Our previous study indicated that aspirin inhibited breast cancer metastasis by attenuating the Wnt/β-catenin pathway and suppressing FMOD expression by inhibiting the HDAC6 deacetylation of β-catenin [22]. In addition, FMOD is critical for breast cancer cell migration and invasion and is positively regulated by the β-catenin/TCF4/LEF1 complex. Overall, FMOD could be the next therapeutic target in cancer, and the molecules and signaling pathways regulating cancer progression involved in FMOD need further research [22].

In this study, we found that FMOD expression was significantly improved in CRC and was associated with its progression. Furthermore, we screened a 12-mer peptide, RP4, targeting FMOD after five rounds of biopanning, and it had a specific blocking ability for FMOD. Based on in vitro and in vivo results, RP4 directly inhibited the growth and metastasis of CRC cell lines, including HCT116, LoVo, and CT26, through blocking the AKT and Wnt/β-catenin signaling pathways.

## 2. Materials and Methods

### 2.1. Cell Culture and Cell Lines

Wild-type cell lines, including LoVo, HCT116, CT26, and 293T cells, were purchased from the Chinese Academy of Sciences Cell Bank in Shanghai, China. HCT116 and CT26 cells were cultured in Roswell Park Memorial Institute (RPMI) 1640 medium (Gibco, New York, NY, USA) containing 10% fetal bovine serum (FBS) (Gibco, New York, NY, USA), streptomycin (100 µg/mL), and penicillin (100 U/mL). LoVo and 293T cells were cultured in Dulbecco’s modified Eagle’s medium (DMEM) (Gibco, New York, NY, USA) supplemented with 10% FBS, streptomycin (100 µg/mL), and penicillin (100 U/mL) (Gibco, New York, NY, USA). The cells were then incubated in a humidified incubator at 37 °C and 5% CO_2_.

### 2.2. Cell Viability Assay

The MTT assay (Aladdin, Shanghai, China) was conducted to detect the viability of different CRC cell lines. Cells were seeded into 96-well plates at 7.5 × 10^3^ cells/well and cultured in an incubator. After 24 h, gradient concentrations of RP4 with 5 μM, 10 μM, 20 μM, 40 μM, 80 μM, and 160 μM were added to the planking cells, which were then cultured for 24 h, 48 h, and 72 h. After treatment, fresh medium supplemented with 5 mg/mL MTT was added to obtain formazan crystals. Next, 150 µL of Dimethyl sulfoxide (DMSO) (Sigma Aldrich, St. Louis, MO, USA) was added to dissolve the formazan crystals. Absorbance at 570 nm was measured using an EnSight Multimode Plate Reader (PerkinElmer, Singapore).

### 2.3. Phage Display Peptide Library Biopanning

The Ph.D.-12 Phage Display Peptide Library (E8110S, NEB, Ipswich, MA, USA) with a capacity of 1 × 10^9^ was used for biopanning FMOD-binding phage clones according to the manufacturer’s instructions. Briefly, 2.5 × 10^5^ 293T-FMOD^+/+^ cells were produced by transduction of lentivirus carrying FMOD ORF and were seeded into 6-well plates. 293T wild-type cells were transduced with the lentivirus pcDNA vector as control. After blocking, FMOD-binding phage clones were enriched by panning the phage library using the subtractive screening method [23]. Briefly, 10 µL of the phage library kit in 2 mL DEME was added to 293T-pcDNA for 1 h to remove non-specific phage clones. Subsequently, the supernatant was added to the 293T-FMOD^+/+^ cells and incubated for another 2 h. After removing the supernatant, the phage library binding to 293T-FMOD^+/+^ cells was eluted with 0.1 M glycine–HCl (pH 2.2) and neutralized with 1 M Tris–HCl (pH 9.1). The titration of the biopanning phage was measured by transfecting *E. coli ER2738* in the exponential phase and plating on Luria Bertani agar plates containing tetracycline overnight at 37 °C. Overall, five rounds of biopanning were performed to enrich FMOD-specific binding phage clones.

### 2.4. Molecular Docking Studies

The molecular docking studies were completed by Surflex-Dock module of Tripos SYBYL software. Firstly, we defined the dummy atoms and parameters for the metal atoms and applied various force field constraints into calculations. Then, a dummy atom for metal atoms in complexes was utilized in analyzing the binding affinities and binding sites of FMOD protein with RP1, RP2, RP3, RP4, and RP5, respectively.

### 2.5. Peptide Synthesis

Candidate peptides RP1, RP2, and RP4 were synthesized by China Peptides (Shanghai, China) by normal solid-phase Fmoc chemistry. RP4 labeling with FITC was referred to the instructions of FITC Labeling Kit (#53027, Pierce, Rockford, IL, USA) [24]. Briefly, 40 μL of the borate buffer (0.67 M, pH 8.5) and 0.5 mL of 2 mg/mL peptide in PBS and FITC reagent were mixed thoroughly. Then the mixture was incubated in the darkness for 2 h at room temperature. The labeling solution was applied onto Sephadex-G25 column and the labeled protein (yellowish color) was pooled and stored at −80 °C. The synthesized peptides were validated by reverse-phase HPLC and mass spectrometry analysis.

### 2.6. HPLC and MS Analysis

For the High Performance Liquid Chromatography analysis, 1 mg of sample were dissolved in 1 mL of water (or response solvent), and were shaken by ultrasound until clear and transparent. Then, 10 µL of peptide sample were used for analysis. For the mass spectrometry, 1 mg sample was dissolved in 1 mL of water (or solvent of the appropriate proportion) and shaken by ultrasound until clear and transparent. The sample was delivered to the injection vial and analyzed by electrospray ionization (ESI) to obtain an MS spectrum.

### 2.7. Immunofluorescence

CRC cells were seeded into confocal dishes and cultured until 70% confluence was reached. The cells were fixed with paraformaldehyde for 15 min and incubated with primary antibodies against FMOD overnight at 4 °C. After washing with PBS, the cells were incubated with secondary antibodies and the RP4-FITC peptide. Finally, DAPI was used to stain the cell nuclei, and images were captured using a super-resolution microscope (Nikon, New York, NY, USA).

### 2.8. Cell Apoptosis Analysis

CRC cells (4 × 10^5^) were seeded in 6-well plates and treated with RP4 at concentration of 0 μM, 50 μM, 100 μM, and 200 μM for 48 h. After digestion with 0.05% trypsin, cells were collected and stained with the Annexin V-FITC/PI Kit (4A Biotech, Suzhou, China) for apoptosis detection, according to the manufacturer’s protocol. Apoptosis was analyzed using the FlowJo software (Beckman, California, CA, USA).

### 2.9. Western Blot

Cells treated with RP4 at concentrations of 0 μM, 50 μM, 100 μM, and 200 μM were collected and lysed using a radioimmunoprecipitation assay (RIPA). Next, the total protein concentration was measured using the BCA assay to ensure a unanimous loading volume. The primary antibodies used were caspase 3 (A19664), caspase 9 (A19664), E-cadherin (A3044), N-cadherin (A19083), phospho-AKT (AP0637), Cyclin E2 (A9305), β-catenin (A19657), phospho-β-catenin (AP0579), Snail (A5243), Phospho-CCND1 (AP1061), vimentin (A11952), cyclin D1 (A19038), Bcl2 (A111025), Phospho-ERK1 (AP0472), and ERK1/ERK2 (A10613). All of the above were purchased from ABclonal, Wuhan, China. The results were visualized using a chemiluminescent Western blot detection kit (4A Biotech, Suzhou, China).

### 2.10. Cell Migration and Invasion

For cell migration, 2 × 10^5^ CRC cells with gradient concentrations of RP4 at concentrations of 0 μM, 10 μM, 20 μM, and 40 μM were seeded into each Transwell chamber, and the underlayer was supplemented with complementary medium supplemented with 20% FBS. For the invasion assay, Transwell chambers were coated with Matrigel (Corning, New York, NY, USA) in advance to mimic the cellular matrix. The following steps were the same as those used in the cell migration assay. Migrating and invasive cells were visualized using 0.5% crystal violet and captured using an optical inverted microscope (Nikon, Tokyo, Japan).

### 2.11. Angiogenesis Assay

Human umbilical vein endothelial cells (HUVEC) (Chinese Academy of Sciences Cell Bank, Shanghai, China) were seeded into a 96-well plate coated with Matrigel (Corning, New York, NY, USA) and incubated for 6 h with varying concentrations of RP4 at concentrations of 0 μM, 10 μM, 20 μM, and 40 μM. Cells were then stained with 2 ug/mL Calcein AM (Yeasen, Shanghai, China) for 30 min, according to the manufacturer’s instructions. Images were captured using a Nikon microscope, and tube formation ability was analyzed using ImageJ.

### 2.12. Animal Model

Six-week-old female nude BALB/c mice purchased from Guangdong Medical Laboratory Animal Center in China were randomly divided into three groups and subcutaneously injected with HCT116-lucifer cells (5 × 10^6^ cells in 100 μL PBS/mouse) into the left axilla. The animal study was reviewed and approved by the Administrative Committee on Animal Research of Shenzhen International Graduate School, Tsinghua University (Ethical Development No. 16, 10 November 2021). On day 3 after tumor cell injection, mice were treated with RP4 peptides daily via intravenous tail injection at concentrations 0 mg/kg, 100 mg/kg, 150 mg/kg, respectively. Body weight and tumor size were measured and recorded every 2 days. Primary tumor volume was calculated using the following formula: (length) × (width) ^2^ × 0.5. Tumor development was monitored twice a week using luciferin with 15 μg/g (Goldbio, St. Louis, MO, USA). After 15 days of treatment, the mice were sacrificed and tumors, blood, and other viscera were collected for further research. Data analysis was performed using Living Image Software (Caliper Life Sciences, Hopkinton, MA, USA).

For the metastatic CRC model, female Balb/c mice were randomly divided into two groups and intravenously injected with CT26-lucifer cells (1 × 10^6^ cells in 100 μL PBS/mouse). Once lung metastasis was observed, mice were treated with RP4 peptides at the concentration of 100 mg/kg once a day for 20 days. Tumor development was monitored twice a week using luciferin with 15 μg/g (Goldbio, St. Louis, MO, USA). Mice were sacrificed, and blood was collected for further analysis. The lungs and spleen were extracted for flow cytometry analysis.

### 2.13. HE and IHC

The tissues were fixed in the 4% paraformaldehyde for 48 h at room temperature, and then were dehydrated through 75%, 85%, 95%, and 100% ethanol. After being treated by the xylene, the samples were embedded in paraffin. The samples were sliced into 4 µm, which were disposed by the standard process.

The human clinic colorectal tissue and adjacent sections were stained with FMOD antibodies (A6375) after deparaffinization, rehydration, and antigen repair. The mouse tissues were stained with Ki67 (A2094) and Bcl2 antibodies (A111025). All of the above were purchased from ABclonal, Wuhan, China. Then the samples were treated with corresponding secondary antibody at room temperature for 1 h. The samples were stained with hematoxylin. Finally, the images were captured under a microscope (Leica, Wetzlar, Germany).

### 2.14. In Vivo Biodistribution

The RP4 distribution in vivo were measured as previously described [25]. The female nude BALB/c mice with tumors were randomly divided into three groups with five in each group. The PBS, 100 mg/kg RP4 and 150 mg/kg RP4 (labeled with His-tag) were injected through the caudal vein. Then peripheral blood was collected at 3 h, 6 h, 12 h, and 24 h, respectively, and the tumors and organs were collected after 24 h. The RP4 concentration were analyzed by Human His-Tag ELISA Kit (Shenzhen Ziker Biological Technology, Shenzen, China).

### 2.15. Flow Cytometric Analysis

Mouse lungs and spleens were obtained to generate single-cell suspensions for flow cytometric analysis. Briefly, the lungs were extracted, mildly minced into small pieces, and dissociated using digestive enzymes. Furthermore, a 70-µm cell strainer was used to insulate single cells from the tissue matrix. Single-cell suspensions were washed with DMEM complete medium and lysed with red blood cells (RBC) lysis solution (Miltenyi Biotec, Bergisch Gladbach, Germany). Finally, the single-cell suspension was collected by centrifugation at 300× *g* for 10 min.

Single-cell suspensions isolated from the spleens were generated using the mechanical digestion method. Spleens with no connective tissue or fat were extracted from a slaughtered mouse. The flat end of the plunger from a sterile 3 cc syringe was used to crush the spleen in gentle circular motions to release the splenocytes. The mixture was then collected into a sterile 50 mL conical tube by gently grinding the spleen tissue through a 70-µm cell strainer (Miltenyi Biotec, Gaithersburg, MD, USA) to dissociate single cells. After centrifuging the tube at 300× *g* for 10 min, the RBC lysis solution was used again to remove the RBC.

For cell gating, dead cells were excluded using live/dead staining. CD3 (BioLegend, San Diego, CA, USA) was used as a marker for total leukocytes, mouse Treg cell subset was identified by markers CD4 (TONBO, San Diego, CA), CD25 (BioLegend, San Diego, CA, USA), and Foxp3 (TONBO, San Diego, CA, USA). Mouse CD8^+^ T cells were collected as CD3^+^CD8^+^Granzyme-B (FITC, BioLegend, Cat. No. 100203; PerCP.Cyanine5.5; TONBO, Cat. no. 65-0081-U025; PE, BioLegend, Cat. no. 372207), and natural killer T (NKT) cells were gated as CD3^+^CD8^+^CD49b (FITC, BioLegend, Cat. No. 100203; PerCP.Cyanine5.5; TONBO, Cat. No. 65-0081-U025; APC/Cyanine7, BioLegend, Cat. No. 108919). The stained cells were detected using a Beckman flow cytometer, and the data were analyzed using FlowJo Software (Beckman, California, CA, USA).

### 2.16. Toxicology Assays

The orbital blood samples were collected into 1.5 mL microcentrifuge tubes when the mice were sacrificed, and store samples were kept at 4 °C for 2 h or overnight. The samples were centrifuged at 10,000× *g* for 10 min at 4 °C, the supernatant was collected and the cell pellet was discarded. Repeat the centrifugation and store the serum at −20 °C. The serum was analyzed by the Alanine Aminotransferase (ALT), triglyceride (TG), and γ-glutamyltransferase (γ-GT) kits (Jinmei Biotech, Shandong, China).

### 2.17. Statistical Analysis

All experiments were repeated a minimum of three times, and the data are presented as mean ± SD. All data were analyzed using Prism 8.0 (GraphPad, California, CA, USA) software. The normality tests for in vivo data were finished with Shapiro–Wilk test. Statistical differences between data groups were evaluated for statistical significance using the t-test (comparing two experimental groups) for unpaired data or one-way ANOVA (more than two groups involved). *p* was set to 0.05.

### 2.18. Data Availability

All the data needed to evaluate the conclusions of the study are presented in this paper. Additional data related to this study may be requested from the corresponding author.

## 3. Results

### 3.1. Expression Status of FMOD Is Related to CRC Progress

To further explore the relationship between FMOD and CRC, we compared FMOD levels in CRC. We found that the protein levels of FMOD were significantly elevated in human CRC cell lines, including HCT116, LoVo, and Caco_2_ (Figure 1A). As shown in Figure 1B, human CRC tissues had a higher expression of FMOD than the adjacent tissues that are more than 2 cm away from the tumor edge [26]. Interestingly, FMOD expression in CRC patients was significantly associated with poor prognosis (Figure 1C). To examine whether FMOD regulates the immune microenvironment during CRC development, we utilized TIMER2.0 to analyze the relationship between FMOD expression and immune cells in CRC. The results revealed that FMOD was significantly negatively correlated with CD8^+^ T cells and NKT cells in CRC (Figure 1D). According to single-cell RNA sequencing data from a single-cell portal (https://singlecell.broadinstitute.org/ (accessed on 28 October 2021)), epithelial and stromal cells had the highest expression levels of FMOD in CRC (Figure 1E(a,b)). Taken together, these data suggested that FMOD had a clinically significant association with CRC progression and may promote CRC development by regulating tumor-associated immune cells.

### 3.2. Biopanning of FMOD Antagonist Peptides

To further investigate the role of FMOD in CRC, we evaluated a Ph.D.-12 phage display peptide library in HEK293T FMOD^+/+^ cells to identify peptides that bind to FMOD (Figure 2A). Five phage clones were identified after five rounds of biopanning. To further clarify the binding ability of peptides to FMOD, we used SYBYL 8.0 software (Tripos, St. Louis, MO, USA) to simulate the binding of the five peptides and FMOD. Theoretically, the docking scores and hydrophilicity characteristics of RP4 were better than those of the other four peptides, suggesting that RP4 may bind to FMOD more effectively (Figure 2B). As shown in Figure 2C, the molecular docking model showed that RP4 binds to FMOD through ASN-193, TYR-257, ARG-236, ARG-190, LYS-237, ASP-240, VAL-238, SER-262, GLY-266, and THR-258. Therefore, RP4 was selected for further study of its effects on CRC.

### 3.3. RP4 Binds to FMOD and Inhibits the Growth of Colorectal Cancer Cells

Based on the molecular docking prediction, RP1, RP2, and RP4 were selected for synthesis. To compare the anti-cancer activities of the three peptides, an MTT assay was conducted. As shown in Figure 3Aa, RP4 displayed more effective inhibition of cell viability of 293T FMOD^+/+^ cells, with an effective IC_50_ of 93.31 ± 0.38 μM. We analyzed the physical properties of RP4, and the results showed that RP4 is approximately hydrophilic to FMOD (Appendix A). High-performance liquid chromatography (HPLC) (Appendix A) and mass spectrometry (MS) (Appendix A) were used to confirm 98% purity of the RP4 peptide. The MTT assay was performed to assess the effects of RP4 on FMOD-overexpressing CRC cell lines. RP4 significantly suppressed the growth of LoVo, HCT116, and CT26 cells in a dose-dependent manner (Figure 3Ab). To visualize the combined status of RP4 and CRC cell lines, FITC was labeled on RP4 peptides and used in the immunofluorescence assay. We also compared FMOD antibody and RP4 in inhibiting cell viability of HCT116 and CT26 shown in Appendix A and 2B. In vitro, RP4 has better effect in suppressing human colorectal cancer cell proliferation with a lower IC50 value when compared with FMOD antibody. As shown in Figure 3B, RP4 bound to FMOD-overexpressing CRC cell lines, including LoVo, HCT116, and CT26, but not to Caco2 and 293T cells, which are FMOD low expressing cell lines_._ Furthermore, an annexin V/PI flow cytometry assay was used to determine whether RP4 could induce apoptosis in CRC cells. Flow cytometry data showed that RP4 significantly induced apoptosis in HCT116 and CT26 cells (Figure 3C). In addition, Western blotting results showed that RP4 treatment with concentrations up to 200 uM led to decreased levels of p-Akt, p-ERK, and Bcl-2. In contrast, protein levels of Bax and cleaved caspase-3/9 were significantly elevated (Figure 3D). In Appendix A, lower concentrations of RP4 consistent with Figure 3D also suppressed AKT and ERK phosphorylation and increased expression of Bax and cleaved caspase-3/9 in HCT116 cells. Overall, these data implied that RP4 effectively bound to FMOD and inhibited CRC cell growth via caspase-3-induced apoptosis.

### 3.4. RP4 Inhibits CRC Cells Invasion and Migration via Suppressing the Wnt/β-catenin Signaling Pathway and Blocking Angiogenesis of HUVECs

To determine the inhibitory effects of RP4 on the invasion and migration of CRC cells, a Transwell assay was performed. RP4 inhibited the migration and invasion of LoVo, HCT116, and CT26 CRC cells in a dose-dependent manner (Figure 4A,B). To further evaluate the effects of RP4 on capillary tube formation activity of HUVECs, HUVECs were treated with various concentrations of RP4. As shown in Figure 4C, RP4 significantly reduced the total number of endpoints, average vessel length, total number of junctions, and junction density during the tube formation. As shown in Figure 3D, Western blotting results showed that RP4 treatment inhibited the phosphorylation of AKT, a crucial positive regulator of the oncogenic signaling mechanism. The deactivation of AKT led to elevated phosphorylated β-catenin, indirectly inducing a decrease in N-cadherin, MMP9, and snail levels while increasing E-cadherin levels in CRC cells, resulting in inhibition of the EMT process (Figure 4D). We also found that the ubiquitination and acetylation levels of β-catenin were significantly increased in RP4-treated cells, suggesting that RP4 may induce β-catenin degradation. Overall, the results suggested that RP4 inhibited CRC cell migration and invasion by inhibiting the EMT process via suppression of the AKT and Wnt/β-catenin signaling pathways.

### 3.5. RP4 Suppress CRC Xenografts Growth In Vivo

The therapeutic efficacy of RP4 in vivo was assessed in a xenograft tumor model, which was constructed by implanting HCT116-luciferase cells into BALB/c nude mice. Tumor growth was monitored by luciferase bioluminescence. As shown in Figure 5A–D, RP4 treatment suppressed tumor growth in terms of both weight and volume. The IHC results showed that RP4 significantly decreased the levels of Ki67 and Bcl2, indicating that tumor cell proliferation was inhibited by RP4 (Figure 5E). The immunofluorescence of TUNEL significantly increased after treatment with RP4 in a dose-dependent manner (Figure 5F). The in vivo biodistribution result in Figure 5H suggested that RP4 has significant targeted ability to tumors compared to other organs including heart, liver, lung, spleen, and kidney. In Figure 5G, it shown that RP4 in serum peaked in 3 h, and lasted around 24 h in vivo. To test the toxicity of RP4 in vivo, HE staining and ELISA for liver function test indices were performed, and these data showed that there was no obvious toxicity in the RP4-treated group (Appendix A). Taken together, these data indicated that RP4 effectively inhibited CRC growth in vivo without obvious toxicity.

### 3.6. RP4 Inhibits CRC Lung Metastasis In Vivo by Reprogramming Immune Cell Activities

To assess the effect of RP4 on the lung metastasis ability of CRC in vivo, CT26-luciferase cells were injected into Balb/c mice to generate the lung metastasis model. Continuous RP4 treatment significantly suppressed the metastasis of CT26 cells in the lung tissue (Figure 6A,B), without obvious weight loss (Figure 6C). After 2 weeks, the mice were sacrificed to obtain lung tissues, and representative images are shown in Figure 6D,E. Pulmonary metastatic nodules were significantly decreased in the RP4-treated group (Figure 6F). Taken together, RP4 significantly suppressed the formation of CRC metastatic lung nodes.

Immune cells are essential components of the TME because of their role in monitoring the immune microenvironment [27]. It has been widely reported that the Treg cell population works as a protagonist in immune suppression by releasing suppressor cytokines such as IL-10 and TGF-β, or by depriving IL-2 [28,29]. The expression of Foxp3 and CD25 is necessary for the stable immunosuppressive activity of Tregs. Previous clinical data implied that Treg cells are positively correlated with CRC progression [30]. In breast metastatic cancer, Tregs may promote the progression of lung metastasis by regulating NK cells [31]. CD8^+^ cytotoxic T lymphocytes (CTLs) are the preferred immune cells for inhibiting cancer cell growth, but they encounter dysfunction and exhaustion during cancer progression [32]. In addition, the immunosuppressive population, including Treg cells, hinders CD8^+^ T cell-mediated antitumor immune responses to promote cancer cell survival [33]. Therefore, it is necessary to activate CD8^+^ T cells to enhance anti-tumor immune reactions and produce durable and efficient antitumor immune responses. Unlike conventional T cells, the NKT cell population constitutes a unique subset of T cells that interface between innate and adaptive immunity and possesses cytotoxic capabilities in cancer progression [34]. As shown in Figure 7A,B, RP4 treatment significantly decreased the level of CD25^+^Foxp3^+^Treg cells in both the lung and spleen lymph nodes, whereas it increased CD8^+^ T cell and NKT levels, which suggested that RP4 regulated immune cells in metastatic sites and lymph nodes. Moreover, cytokines are necessary in the tumor microenvironment to promote cancer cell survival by suppressing the immune response. The classic cytokines that promote tumorigenesis include IL1β, IL2, IL10, TGF-β, and TNF-α. Conversely, IFN-γ strongly stimulates the antitumor immune response. As shown in Figure 7C, RP4 dramatically decreased the levels of IL-1β, IL-2, IL-10, TGF-β, and TNF-α in the serum, but increased the level of IFN-γ. Overall, these results suggested that RP4 treatment stimulated the anti-cancer immune response to suppress cancer cell growth and metastasis.

## 4. Discussion

Current evidences suggest that FMOD plays many roles in cancer development, including regulation of cancer cell apoptosis [20], promotion of angiogenesis [35] and migration [36]. FMOD is utilized as a biomarker to be evaluated in clinical samples [21], and targeting FMOD can be developed as a useful therapeutic approach for cancer. Our previous study showed that β-catenin translocated into the nucleus to form a transcription complex with TCF4 and LEF1, which then promotes FMOD transcription, subsequently leading to breast cancer cell migration and invasion. Aspirin inhibits breast cancer metastasis by suppressing FMOD expression via the Wnt/β-catenin pathway [22]. According to the single-nucleus RNA sequencing profile of CRC patients, we found that epithelial and stromal cells had high FMOD expression levels in CRC tissues, mainly in goblet cells. This result also implies that FMOD expression is significantly negatively correlated with the infiltration levels of CD8^+^ T cell and NKT cells. We screened the binding of the antagonist peptide RP4 to FMOD using phage display technology for CRC therapy. The RP4 peptide had a better effect on suppressing cell proliferation than FMOD antibody both in CT26 and HCT116 cell. All the data in this study suggest that RP4 significantly represses CRC cell growth, migration, and metastasis both in vitro and in vivo, thus implying that FMOD is a potential target for CRC, and RP4 is a promising drug candidate for the clinic.

Phage display technology is a powerful tool in cancer therapy because phage libraries are widely applied in screening receptor-specific targeting peptides, which may be either agonists or antagonists [37]. Peptide-based drugs are characterized by their low toxicity to normal tissues, low molecular weight, and high specific binding ability to target proteins [38]. Owing to their advantages, including easy availability and convenient purification and storage, peptide-based therapies are widely applied in diverse diseases, such as fibrosis and autoimmune diseases [39]. Peptide-based drugs play various roles in cancer, including early diagnosis, prognostic prediction, and treatment [40]. In particular, in cancer treatment, peptides have shown great potential in both in vitro and in vivo experimental models. Atrial natriuretic peptide, a type of cardiac- and vascular-derived peptide hormone, has been proven to be a potential drug for CRC owing to its anti-proliferative effect via inhibition of the Wnt/β-catenin signaling cascade and increasing intracellular acidification [41]. Another peptide, P56, is isolated from the phage display peptide library, bound to the highly expressed Flt-1 receptor (VEGFR-1) of tumor vascular EC with high affinity and specificity, and significantly inhibited tumor growth and lung metastases [42,43]. Similarly, phage display technology can be used to explore CRC treatments. According to clinical data, FMOD is highly expressed in CRC patients. The screened peptide RP4 suppressed CRC growth by inducing apoptosis and inhibiting cell migration and invasion in vitro and in vivo, suggesting that it is a promising therapeutic tool for CRC.

Apoptosis is programmed cell death that normally occurs during development and morphogenesis. Apoptosis is tightly induced by two core pathways, the extrinsic and intrinsic pathways, which lead to the same terminal [44]. In the intrinsic pathway, the final stage of apoptosis is initiated by caspase 9 and executed by caspase 3, followed by the activation of cytoplasmic endonuclease, leading to cell death [45]. Cancer cells upregulate the anti-apoptotic protein Bcl2 binding to the pro-apoptotic member Bax to restrain pore formation and cytochrome c release in the mitochondria [46]. In the present study, RP4 treatment increased Bax level while decreasing Bcl2 level in CRC cells, along with the activation of caspase 3 and caspase 9, and finally induced CRC cell apoptosis. EMT is a biological process that characterizes the transformation of epithelial cells that interact with the basement membrane into a mesenchymal cell phenotype marked by migratory and invasive capabilities [47]. Many studies have reported that EMT is crucial in the cancer-related metastatic cascade, significantly improving the migratory and invasive capabilities of cancer cells [48]. Current evidence suggests that the Wnt/β-catenin signaling pathway promotes cell migration and invasion by activating the EMT process [49]. In our study, blocking FMOD by RP4 inhibited AKT phosphorylation. As a necessary pivot, AKT interlaces with the Wnt/β-catenin signaling pathway to regulate EMT. When AKT is activated, GSK3β is dephosphorylated and dephosphorylates β-catenin to its active form, which translocates into the nucleus and triggers EMT. In this study, RP4 treatment significantly suppressed the Wnt signaling pathway by repressing AKT to further inhibit the EMT process.

The TME is a complex structure that comprises a wide variety of cell types, including immune cells. Crosstalk between immune cells and cancer cells facilitates tumor cell proliferation, invasion, and metastatic dissemination [50]. T-cell families are essential members of the TME and have a dual influence on cancer cells. CD8^+^ T and NKT cells play key roles in the regulation of antitumor immunity. CD8^+^ T cells are key immune cells that fight against cancer cells, presenting major histocompatibility complex (MHC) class I molecules [51]. NKT cells, with characteristics of both conventional T cells and natural killer (NK) cells, kill cancer cells directly or indirectly through cytokine production [34]. In contrast, Tregs exert their immunosuppressive function in cancer development via various mechanisms, such as the production of immunosuppressive cytokines and immunosuppressive metabolites [52]. RP4 treatment boosted immune cells via increased CD8^+^ T and NKT cells in lung metastasis sites and spleen lymph nodes, and decreased Treg cells. FMOD directly interacts with TGF-β by regulating the action of local TGF-β within ECM, and works as a crucial regulator of glioma cell migration in the downstream of TGF-β1 pathway [36,53]. Accumulating evidence has found the TGF-β suppress tumor immune microenvironment by promoting Treg cells proliferation and immunosuppressive function. Additionally, TGF-β suppress CTL and NK cells function to accelerate tumor progress [54,55]. Our results suggested that RP4 promoted immune cells infiltration in lung metastasis sites by blocking FMOD function and further might inhibit the impact of TGF-βin tumor progress, which needs future study. In addition, the cytokines produced in the TME play an important role in cancer pathogenesis. RP4 treatment decreased IL-1β, IL2, IL10, TGF-β, and TNF-α levels in the serum and increased IFN-γ levels. These results suggested that RP4 of FMOD significantly boosts the immune system to repress CRC development in vivo.

In summary, we screened and identified a novel FMOD antagonist peptide, RP4, from a phage display library, and demonstrated its ability to suppress CRC progression by inhibiting cancer cell growth and metastasis. RP4 suppressed CRC cell growth by promoting apoptosis and inhibiting CRC cell migration and invasion by repressing the Wnt/β-catenin pathway (Figure 8). Moreover, RP4 can boost the immune system to repress CRC development in vivo by reprogramming related immune cells. Furthermore, our study indicates that FMOD is a potential target and RP4 is a promising therapeutic agent for CRC treatment.

## Figures and Tables

**Figure 1 pharmaceutics-15-00944-f001:**
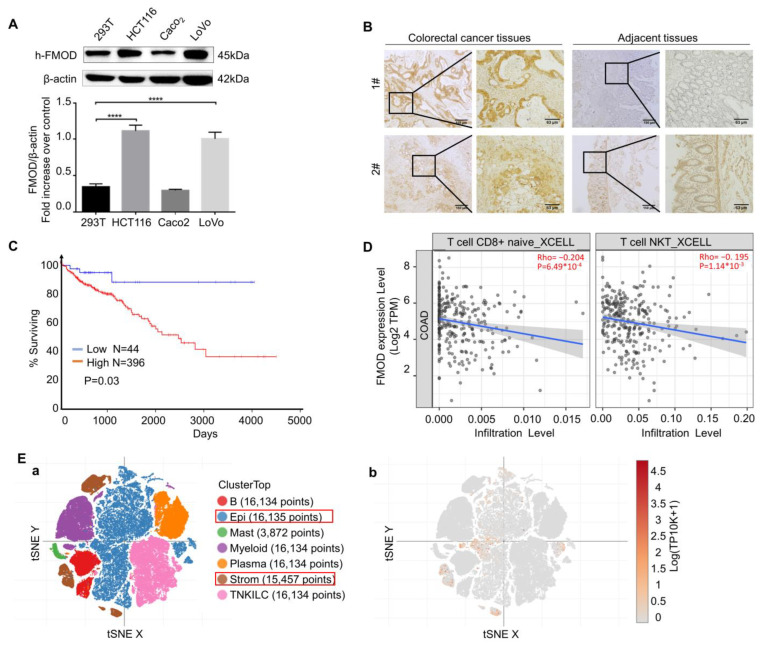
The expression of FMOD is associated with CRC progression. (**A**), Protein expression levels of FMOD in CRC cell lines compared with 293T cells were measured by Western blotting. (**B**), IHC staining of FMOD in CRC and adjacent tissues. Adjacent tissues located more than 2 cm away from the malignant tumor edge and taken out by experienced surgeons. (**C**), Kaplan–Meier survival analysis and log-rank test showed the overall survival of CRC patients who were FMOD-positive (*n* = 398) vs. FMOD-negative (*n* = 44). Data from OncoLnc. (**D**), Correlation between FMOD expression levels and CD8^+^ T cell and NKT cell infiltration levels. (**E**), Single-cell analysis of FMOD in CRC tissues. (**a**) Tissue clusters in colorectal cancer. (**b**), FMOD is mostly expressed in the epithelium and stroma. **** means *p* < 0.0001.

**Figure 2 pharmaceutics-15-00944-f002:**
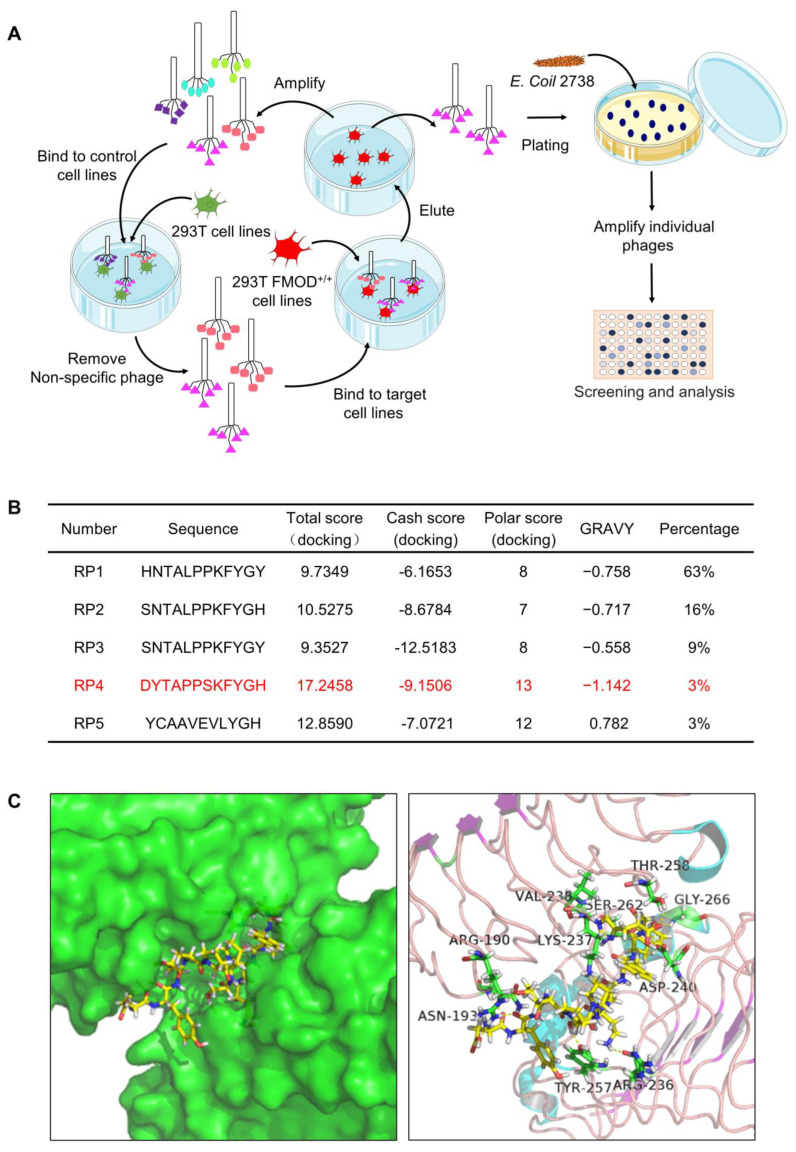
Biopanning process of FMOD antagonist peptides. (**A**), Panning procedure in phage display technology. (**B**), The data of grand average of hydropathy and docking score, including total score, cash score, and polar score. (**C**), Combined simulation of FMOD and RP4 by SYBYL 8.0 software (Tripos, St. Louis, MO, USA).

**Figure 3 pharmaceutics-15-00944-f003:**
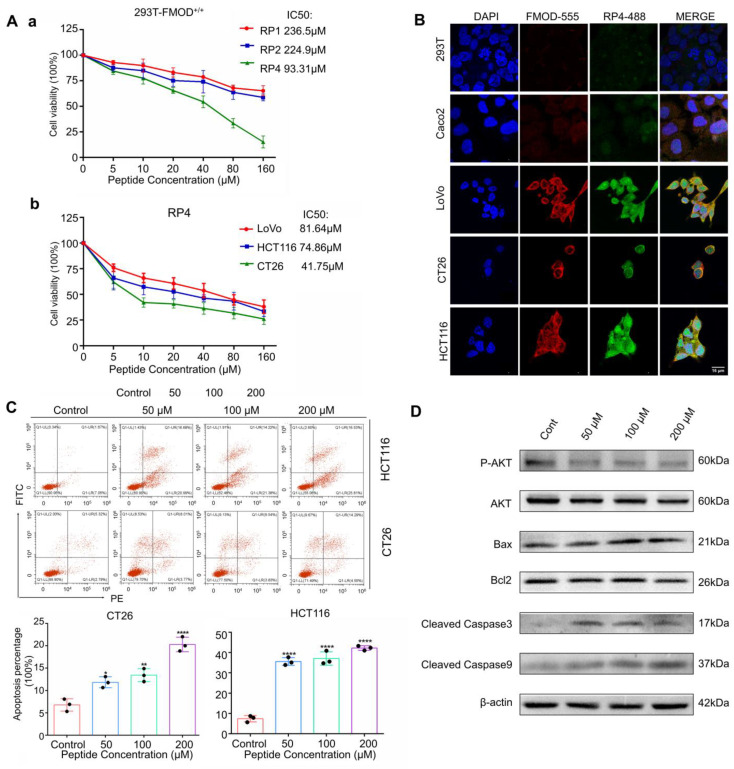
RP4 binds to FMOD and inhibits the growth of CRC cells. (**A**) (**a**), Cell viability of 293T FMOD^+/+^ treated with indicated concentrations of RP1, RP2, and RP4 for 48 h. (**b**), Cell viability of LoVo, HCT116, and CT26 treated with indicated concentrations of RP4 for 48 h. (**B**), Cellular localization of RP4 and FMOD by confocal immunofluorescence. (**C**), HCT116 and CT26 cells were stimulated with indicated concentrations of RP4 for 48 h and then co-stained with PI and FITC conjugated Annexin V. The apoptosis of cells was detected by flow cytometry. (**D**), Western blotting results of CRC cells treated with RP4. Data are expressed as mean ± S.D. Statistical analysis was performed with one-way ANOVA (*n* = 3; * *p* < 0.05, ** *p* < 0.01, **** *p* < 0.0001). Representative images from three independent experiments are shown.

**Figure 4 pharmaceutics-15-00944-f004:**
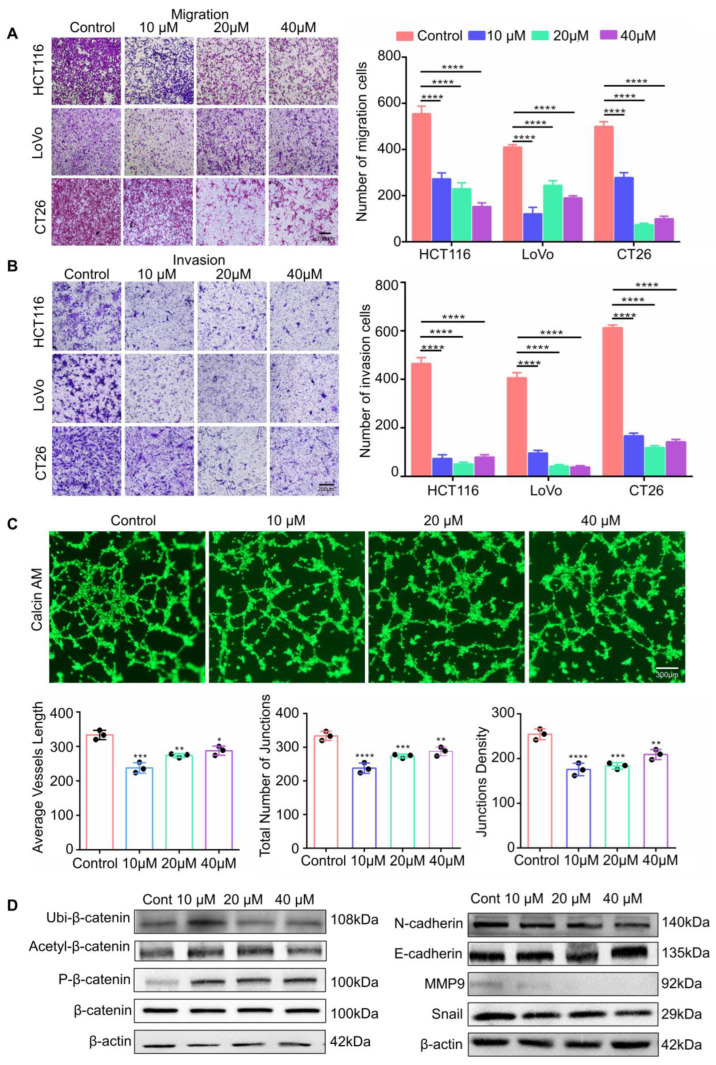
RP4 inhibits invasion and migration of CRC cells and blocks angiogenesis of HUVECs. (**A**,**B**), The migration and invasion assays of HCT116, LoVo, and CT26 cells after RP4 treatment. The migration and invasion of cells were quantified by counting the number of migrating or invasive cells in five randomly chosen fields. (**C**), Tube formation activity of HUVECs cultured in varying concentrations of RP4. Tube formation abilities were measured via the average vessels length, total number of junctions, and junction density. (**D**), Western blot results of CRC cells treated with RP4. Data are expressed as mean ± S.D. Statistical analysis was performed with one-way ANOVA (*n* = 3; * *p* < 0.05, ** *p* < 0.01, *** *p* < 0.001, **** *p* < 0.0001). Representative images from three independent experiments are shown.

**Figure 5 pharmaceutics-15-00944-f005:**
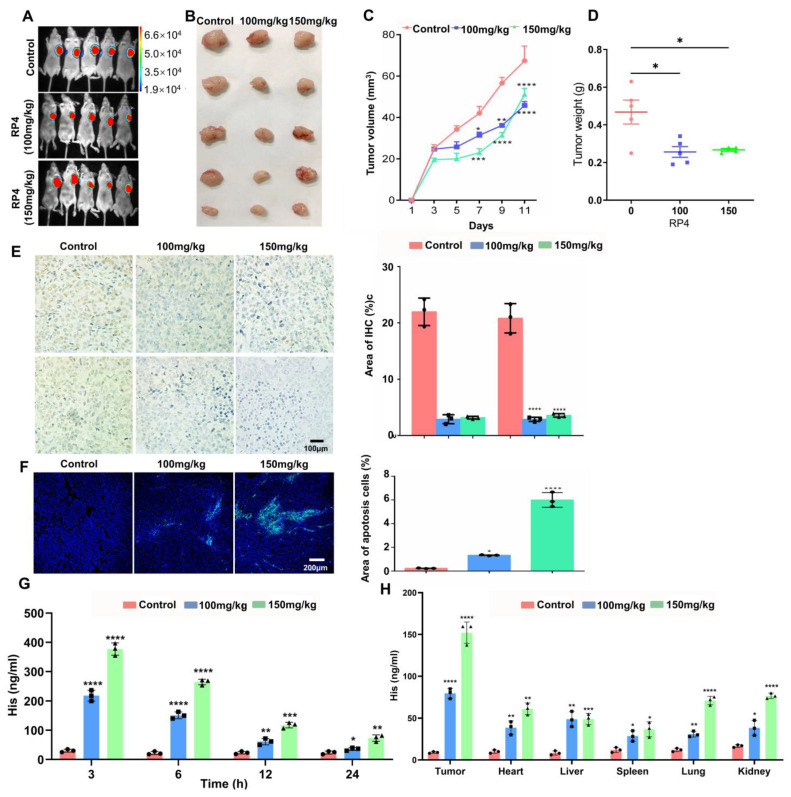
RP4 suppresses CRC xenografts growth in vivo without obvious toxicity. (**A**), Bioluminescence imaging of tumor growth. (**B**–**D**), Tumor weight and volume after RP4 treatment (*n* = 5). (**E**), Immunohistochemistry analysis of Ki67 and Bcl2 expression in tumor. (**F**), RP4 induces apoptosis in tumors, assessed by TUNEL. (**G**), Serum ELISA of xenografted mice treated by PBS, 100mg/kg RP4, 150 mg/kg RP4. (**H**), ELISA of heart, liver, spleen, lung, and kidney treated by PBS, 100mg/kg RP4, 150 mg/kg RP4. Data are expressed as mean ± S.D. Statistical analysis was performed with one-way ANOVA (*n* = 3; * *p* < 0.05, ** *p* < 0.01, *** *p* < 0.001, **** *p* < 0.0001). Representative images from three independent experiments are shown.

**Figure 6 pharmaceutics-15-00944-f006:**
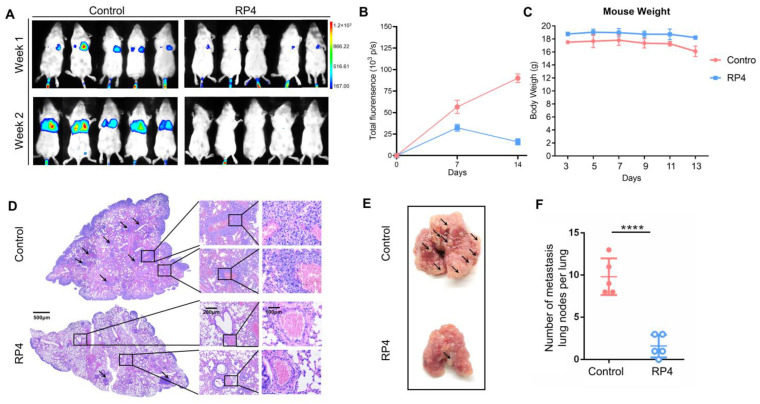
RP4 inhibits colorectal cancer lung metastasis in vivo. (**A**), Bioluminescence imaging of CT26 tumor-bearing mice injected with luciferase substrate. (**B**), Body weight curves (*n* = 6). (**C**), Representative H&E stain showing the metastases area of the lung from CT26 tumor model. (**D**), Representative images of lung metastases in a CT26 tumor model. Arrows represent pulmonary nodules. (**E**), The representative image of lung with metastatic nodes. Arrows represent pulmonary nodules. (**F**), The statistic of metastatic lung nodes (*n* = 6). Data are expressed as mean ± S.D. Statistical analysis was performed with *t* test (*n* = 3; **** *p* < 0.0001). Representative images from three independent experiments are shown.

**Figure 7 pharmaceutics-15-00944-f007:**
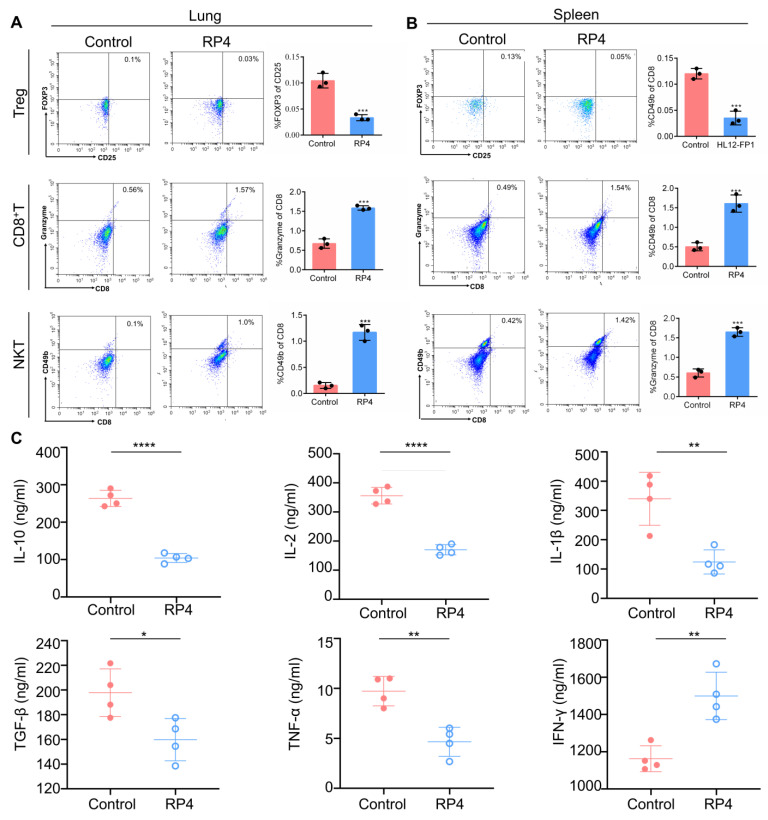
RP4 suppresses lung metastasis by reprogramming the regulation of immune cell activities. (**A**), Flow cytometry analysis of Treg, CD8^+^ T, and NKT cells infiltration in the lung. (**B**), Flow cytometry analysis of Treg, CD8^+^ T, and NKT cells infiltration in the spleen lymph node. (**C**), ELISA analysis of IL-10, IL-2, IL-1β, TNF-α, and TGF-β. Data are expressed as mean ± S.D. Statistical analysis was performed with *t* test (*n* = 3; * *p* < 0.05, ** *p* < 0.01, *** *p* < 0.001, **** *p* < 0.0001). Representative images from three independent experiments are shown.

**Figure 8 pharmaceutics-15-00944-f008:**
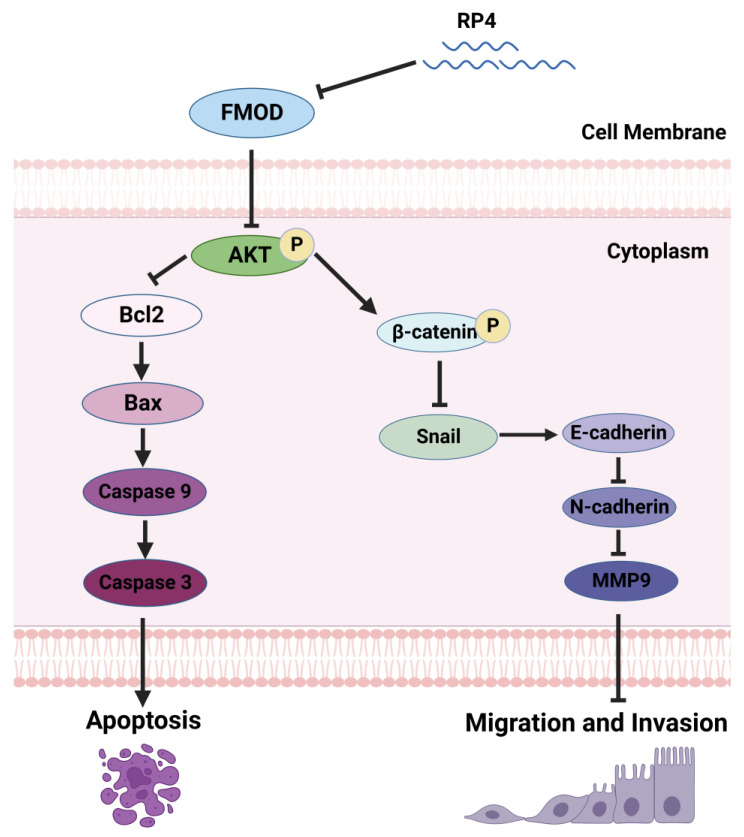
RP4 induces cell apoptosis and inhibits cancer metastasis via AKT and Wnt/β-catenin signaling pathways. RP4 was ingested by colorectal cancer cells and blocked FMOD function, resulting in inhibiting the phosphorylation of AKT and further induing the activation of caspase 9 and phosphorylation of β-catenin. As apoptosis initiator, caspase 9 activated the cleavage of caspase 3, which directly promoted cell apoptosis. In the canonical Wnt/β-catenin pathway, β-catenin phosphorylation led to degradation of β-catenin and reduced the accumulation of β-catenin in the cell nucleus. Thus, β-catenin phosphorylation indirectly inhibited MMP9 and cyclin D1 expression. In addition, blocked FMOD inhibited the phosphorylation of Erk and snail expression. Therefore, down-regulation of MMP9, cyclin D1, and snail repress cell migration or invasion by inhibiting EMT as well as down-regulation of N-cadherin and vimentin.

## Data Availability

Data are available on request.

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
