# Peer review of "A Novel Fibromodulin Antagonist Peptide RP4 Exerts Antitumor Effects on Colorectal Cancer"

_pharmaceutics, 2023, doi:10.3390/pharmaceutics15030944_

Round 1

Reviewer 1 Report

The manuscript by Deng et al. reports a peptide named RP4 that was identified to bind FMOD via phage display and that manifested inhibitory effect on colorectal cancer cell survival and on in vivo tumor formation and progression. The authors described the process of selection of a candidate peptide in detail and analyzed its effect on cancer cell survival and cancer progression in diverse aspects including immune cell profiles and toxicity. The results presented here is interesting enough to draw attention of peers. Here are comments for the authors to address adequately.

1. Source (company, location of the company) of materials used in the experiment should be provided. Antibodies purchased from same company could be described as ‘All of the above from---‘

2. All abbreviation should be described fully at first appearance. For example, DEME at line 110.

3. Caco2 cell line was not described in Section 2.1. In addition, Caco2 cell expressed FMOD lower than the other cells. What would be the effect of RP4 treatment to Caco2 cells (as in lines 299-300)? This result could be one of way to show specificity of RP4 effect on FMOD.

4. Is the expression level of FMOD in CRC patients graded numerically? What would be the cumulative survival rate if numerical grade information on FMOD expression level is considered?

5. The source of ‘single-cell RNA sequencing data’ at line 259 should be provided.

6. Is HEK293 at Line 275 HEK293T?

7. ‘annex/PI’ at line 307 should be ‘annexin V/PI’.

8. Is ‘Fig. 4D’ at line 313 and line 335 ‘Fig. 3D’.

9. What is FMOD expression level in CT26?

10. In Fig. 4C, average vessel length, total number of junctions and junction density seem to increase with increased amount of RP4. Is there any statistical significance?

11. Fig. 5G at line 363 and Fig. 5H at line 365 are reversed.

12. The rationale to use CT26 in metastasis analysis should be provided.

13. In Fig. 5C, in vivo tumor size upon treatment of RP4 at 150 mg/kg seems to catch up that of 100 mg/kg RP4 treatment on day 11. An explanation should be provided.

14. What is the rationale to use 50, 100 and 200 mM of RP4 in survival analysis which are much higher than 10, 20 and 40 mM RP4 in migration and invasion analysis.

15. Is ‘extrinsic pathway’ at line 468 ‘intrinsic pathway’?

16. The notion at lines 484-485 contradicts that at line 337.

17. FMOD is an extracellular protein. Location of FMOD should be correctly indicated in Fig. 8.    

Author Response

Response to Reviewers’ Comments

Dear Editors and Reviewers,

We are grateful to reviewers for taking the time to review our manuscript and give us their valuable comments. We have considered all the comments and have made appropriate changes to the manuscript. Our point-by-point response appears below in blue. The changes made to the main manuscript file were highlighted.

Review 1

Comments to the Author

The manuscript by Deng et al. reports a peptide named RP4 that was identified to bind FMOD via phage display and that manifested inhibitory effect on colorectal cancer cell survival and on in vivo tumor formation and progression. The authors described the process of selection of a candidate peptide in detail and analyzed its effect on cancer cell survival and cancer progression in diverse aspects including immune cell profiles and toxicity. The results presented here is interesting enough to draw attention of peers. Here are comments for the authors to address adequately.

Response: Thanks for your comments, which are really helpful for improving our work. We have corrected the manuscript and made point-by-point responses according to your comments. Kindly please review our re-submit manuscript.

  1. Source (company, location of the company) of materials used in the experiment should be provided. Antibodies purchased from same company could be described as ‘All of the above from---‘

Response: The authors appreciate the reviewer raising these valuable comments. We apologize that the source of materials was missed in the previous submission. We have supplied them following your suggestion.

  1. All abbreviation should be described fully at first appearance. For example, DEME at line 110.

Response: The abbreviations have been fully described

  1. Caco2 cell line was not described in Section 2.1. In addition, Caco2 cell expressed FMOD lower than the other cells. What would be the effect of RP4 treatment to Caco2 cells (as in lines 299-300)? This result could be one of way to show specificity of RP4 effect on FMOD.

Response: Thanks so much for your suggestion. The expression level of FMOD in Caco2 cells is very low when compared with HCT-116, LoVo and CT26 (Figure 3B). Therefore, the Caco2 cell line is used as negative control in Figure 3B. We have treated the Caco2 with RP4 in the preliminary experiment, which showed that the RP4 ineffectually inhibited Caco2 cells growth in MTT assay (Response Figure 1). Thus we did not use the Caco2 in following research. Moreover, the ineffectiveness of FMOD in Caco2 cells suggests that it may be one way of show the specificity of RP4 for FMOD.

Response Figure 1. The inhibition effect of RP4 on the cell proliferation of Caco2 cells.

  1. Is the expression level of FMOD in CRC patients graded numerically? What would be the cumulative survival rate if numerical grade information on FMOD expression level is considered?

Response: Thanks for raising this question. In order to confirm the expression level of FMOD in CRC patients graded numerically and the relationship between CRC survival and FMOD expression, the GEPIA database (http://gepia.cancer-pku.cn/detail.php) was used. The results showed that the expression level of FMOD was gradually increased from stage I to stage III (Response Figure 2A), while slightly decreased in stage IV. Moreover, the high level of FMOD was significantly correlated with poor survival (Response Figure 2B). Taken together, these data suggest that FMOD has a clinically significant association with CRC progression.

Response Figure 2. FMOD expression is upregulated in CRC tissues and predicts a poor prognosis. A, The expression level of FMOD in different stages in CRC. B, CRC patients with high FMOD expression have poor survival. Data was analyzed by GEPIA database (http://gepia.cancer-pku.cn/detail.php).  

  1. The source of ‘single-cell RNA sequencing data’ at line 259 should be provided.

Response: We have added the website of the ‘single-cell RNA sequencing data’ at line 259: https://singlecell.broadinstitute.org/

  1. Is HEK293 at Line 275 HEK293T?

Response: Yes, the HEK293 is HEK293T, and we have corrected it.

  1. ‘annex/PI’ at line 307 should be ‘annexin V/PI’.

Response: Thank you for pointing out the problem, we have corrected it.

  1. Is ‘Fig. 4D’ at line 313 and line 335 ‘Fig. 3D’.

Response: Yes, we are sorry for the mistake and we have corrected them into Fig. 3D

  1. What is FMOD expression level in CT26?

Response: As the immunofluorescent results show in Fig 3B, the FMOD expression is high in CT26.

  1. In Fig. 4C, average vessel length, total number of junctions and junction density seem to increase with increased amount of RP4. Is there any statistical significance?

Response: Actually, there is statistical significance between 10 uM and 40 uM in average vessel length, total number of junctions and junction density, as shown in Response Figure 3. In addition, these results indicated that the 10 uM may have better inhibition effect when compared with 20 uM and 40 uM, and the 20 uM and 40 uM may induce the drug resistance in tube formation activity of HUVECs.

Response Figure 3. Tube formation activity of HUVECs cultured in varying concentrations of RP4. Tube formation abilities were measured via the average vessels length (A), total number of junctions (B), and junction density (C).

  1. 5G at line 363 and Fig. 5H at line 365 are reversed.

Response: We are sorry for the mistake and we have corrected it

  1. The rationale to use CT26 in metastasis analysis should be provided.

Response: CT26 is a murine colorectal carcinoma cell line from a BALB/c mouse [1]. The CT26 CDX model is an allograft (mouse cells implanted in a mouse), thus resulting in the ability to use a fully immune competent mouse. At the same time, CT26 has been successfully utilized for building the lung metastasis model in other publication [2].

  1. In Fig. 5C, in vivo tumor size upon treatment of RP4 at 150 mg/kg seems to catch up that of 100 mg/kg RP4 treatment on day 11. An explanation should be provided.

Response: According to the tumor volume, we found that 150 mg/kg had better inhibition effects of tumor growth before Day 9 when compared with 100 mg/kg. However, long time administration with high dose 150 mg/kg may induce the drug resistance in CRC xenografts model. Therefore, we used 100 mg/kg concentration for following CT26 metastasis model.

  1. What is the rationale to use 50, 100 and 200 mM of RP4 in survival analysis which are much higher than 10, 20 and 40 mM RP4 in migration and invasion analysis.

Response: According to the MTT assay results, the IC50 values of RP4 is exceed 40 uM, which results in half of the cells dead. Thus, in order to confirm the inhibition effect of RP4 in migration and invasion is associated with live cells behavior not because the apoptosis, we used the lower concentrations to keep cancer cells alive.  

  1. Is ‘extrinsic pathway’ at line 468 ‘intrinsic pathway’?

Response: Yes, we are sorry for the mistake and have corrected it

  1. The notion at lines 484-485 contradicts that at line 337.

Response: We are sorry for the mistake and have corrected it.

  1. FMOD is an extracellular protein. Location of FMOD should be correctly indicated in Fig. 8.

Response: We have amended the Fig.8 following your suggestion.

Reference:

  1. Sato, Y., et al., Tumor-immune profiling of CT-26 and Colon 26 syngeneic mouse models reveals mechanism of anti-PD-1 response. BMC cancer, 2021. 21: p. 1-12.
  2. Kee, J.-Y., et al., Gomisin a suppresses colorectal lung metastasis by inducing AMPK/P38-mediated apoptosis and decreasing metastatic abilities of colorectal cancer cells. Frontiers in pharmacology, 2018. 9: p. 986.

Reviewer 2 Report

In the study of Ting Deng et al., it was demonstrated that fibromodulin expression is significantly improved at colorectal cancer and is associated with its progression. A series of peptides, selected based on the molecular doking data, were synthesized and investigated on the ability to inhibite the growth and metastasis of colorectal cancer cell lines (HCT116, LoVo, and CT26) through the AKT and Wnt/β-83 catenin signaling pathways. As the results of this study, fibromodulin was confirmed as a potential target for colorectal cancer treatment, and the novel FMOD antagonist peptide RP4 was proposed as a clinical drug for cancer treatment. The results presented in the article look reasonable and the manuscript may be recommended for publication witn the followong minor remarks:

1. Section 3.3: the authors mention that RP1, RP2, and RP4 compounds were selected for synthesis based on the molecular docking prediction. It is desirable to include information on molecular docking in the Materials and Methods section.

2. Similarly, in the Materials and Methods section, please add information about the details of reverse-phase HPLC and mass spectrometry analysis for synthesized peptides.

Author Response

Response to Reviewers’ Comments

Dear Editors and Reviewers,

We are grateful to reviewers for taking the time to review our manuscript and give us their valuable comments. We have considered all the comments and have made appropriate changes to the manuscript. Our point-by-point response appears below in blue. The changes made to the main manuscript file were highlighted.

Review 2

Comments to the Author

In the study of Ting Deng et al., it was demonstrated that fibromodulin expression is significantly improved at colorectal cancer and is associated with its progression. A series of peptides, selected based on the molecular doking data, were synthesized and investigated on the ability to inhibite the growth and metastasis of colorectal cancer cell lines (HCT116, LoVo, and CT26) through the AKT and Wnt/β-83 catenin signaling pathways. As the results of this study, fibromodulin was confirmed as a potential target for colorectal cancer treatment, and the novel FMOD antagonist peptide RP4 was proposed as a clinical drug for cancer treatment. The results presented in the article look reasonable and the manuscript may be recommended for publication with the following minor remarks:

Response: Thanks for your comments, which are really helpful for improving our work. We have corrected the manuscript and made point-by-point responses according to your comments.

  1. Section 3.3: the authors mention that RP1, RP2, and RP4 compounds were selected for synthesis based on the molecular docking prediction. It is desirable to include information on molecular docking in the Materials and Methods section.

Response: Thanks for pointing out this question, we have added this in Materials and Methods section as following:

2.4 Molecular docking studies

The molecular docking studies were completed by Surflex-Dock module of Tripos SYBYL software. Firstly, we defined the dummy atoms and parameters for the metal atoms and applied various force field constraints into calculations. Then, a dummy atom for metal atoms in complexes was utilized in analyzing the binding affinities and binding sites of FMOD protein with RP1, RP2, RP3, RP4, RP5, respectively.  

  1. Similarly, in the Materials and Methods section, please add information about the details of reverse-phase HPLC and mass spectrometry analysis for synthesized peptides.

Response: Thanks for pointing out this question, we have added this in Materials and Methods section as following:

2.6 HPLC and MS analysis

For the High Performance Liquid Chromatography analysis, 1.0 mg of sample were dissolved in 1 ml of water (or response solvent), and were shaken by ultrasound until clear and transparent. Then 10 µl of peptide sample were used for analysis. For the mass spectrometry, 1.0 mg sample was dissolved in 1 ml of water (or solvent of the appropriate proportion) and shaken by ultrasound until clear and transparent. The sample was delivered to the injection vial and analyzed by Electrospray ionization (ESI) to obtain an MS spectrum.

Reviewer 3 Report

The article presents the data on anticancer effects of novel fibromodulin antagonist peptide RP4. The topic is new and promising. The part of the data on experiments in vitro and molecular simulations is correctly done and well described. As for English, I am not native speaker, for me English is acceptable. Hovewer, the in vivo experiments were wrongly analized.

The major remark:

1) The non-parametric analysis should be used for the data on in vivo experiments.

Minor:

1) In the names of the authors the last sentence is "and". Does it mean that somebody was removed from co-authors?

2) 2.15. Statistical analysis. The non-parametric analysis "Kaplan-Meier survival analysis" (Fig. 1) was not mentioned. The non-parametric analysis should be used for the data on in vivo experiments and added in 2.15.

Author Response

Response to Reviewers’ Comments

Dear Editors and Reviewers,

We are grateful to reviewers for taking the time to review our manuscript and give us their valuable comments. We have considered all the comments and have made appropriate changes to the manuscript. Our point-by-point response appears below in blue. The changes made to the main manuscript file were highlighted.

Reviewer 3

Comments to the Author

The article presents the data on anticancer effects of novel fibromodulin antagonist peptide RP4. The topic is new and promising. The part of the data on experiments in vitro and molecular simulations is correctly done and well described. As for English, I am not native speaker, for me English is acceptable. However, the in vivo experiments were wrongly analyzed.

Response: Thanks for your comments, which are really helpful for improving our work. We have corrected the manuscript and made point-by-point responses according to your comments.

The major remark:

  • The non-parametric analysis should be used for the data on in vivo experiments.

Response: Thanks for your suggestions. Student’s t test generally is used to compare whether there is a statistical difference or difference between two groups, such as comparing the height and weight of two groups, which are generally independent and not related. In vivo experiments, we want to analyze whether there is a statistical difference between control and RP4-treated group (control vs 100 mg/kg or control vs 150mg/kg). Therefore, the Student’s t test was used to analyze statistical difference between two groups. Moreover, recent publications also used ordinary Student’s t tests to analyze the statistical difference [1-3]. We really appreciate your opinion, but ordinary ANOVA or Student’s t tests are also applicable in in vivo analysis.

Minor:

  • In the names of the authors the last sentence is "and". Does it mean that somebody was removed from co-authors?

Response: We are sorry for the mistake and no co-author was removed. We have corrected it at the same time.

2) 2.15. Statistical analysis. The non-parametric analysis "Kaplan-Meier survival analysis" (Fig. 1) was not mentioned. The non-parametric analysis should be used for the data on in vivo experiments and added in 2.15.

Response: Thanks for your suggestions. The response is as the same as the question 1. 

Reference:

  1. Ma, S., et al., YTHDF2 orchestrates tumor-associated macrophage reprogramming and controls antitumor immunity through CD8+ T cells. Nature Immunology, 2023: p. 1-12.
  2. Gao, G., et al., The NFIB/CARM1 partnership is a driver in preclinical models of small cell lung cancer. Nature Communications, 2023. 14(1): p. 363.
  3. Mo, J., et al., Therapeutic targeting the oncogenic driver EWSR1:: FLI1 in Ewing sarcoma through inhibition of the FACT complex. Oncogene, 2022: p. 1-15.

Round 2

Reviewer 3 Report

Dear Authors, the publications in Nature is not a prove of wrongly made analysis for in vivo experiments. To prove the use of parametric ANOVA the normality tests should be done at first (Kolmogorov–Smirnov test,Lilliefors test, Shapiro–Wilk test, etc.). They were not done. Sorry, you are mistaken.

Author Response

Response to Reviewers’ Comments

Dear Editors and Reviewers,

We are grateful to reviewers for taking the time to review our manuscript and give us their valuable comments. We have considered all the comments and have made appropriate changes to the manuscript. Our point-by-point response appears below in blue. The changes made to the main manuscript file were highlighted.

Reviewer 3

Comments to the Author

Dear Authors, the publications in Nature is not a prove of wrongly made analysis for in vivo experiments. To prove the use of parametric ANOVA the normality tests should be done at first (Kolmogorov-Smirnov test, Lilliefors test, Shapiro-Wilk test, etc.). They were not done. Sorry, you are mistaken.

Response: Thanks for your suggestions. We corrected the in vivo analysis by using the Shapiro-Wilk test for normality tests. The test results were included in the Excel sheet as a supplementary information. Besides, the related data analysis description were added in the “Statistical analysis”.

Round 3

Reviewer 3 Report

Please, see comments to Editor.